# Maternal and perinatal mortality and morbidity of uterine rupture and its association with prolonged duration of operation in Ethiopia: A systematic review and meta-analysis

**Melaku Desta** [1]*, **Getachew Mullu Kassa**[1], **Temesgen Getaneh**[1], **Yewbmirt Sharew**[1], **Addisu Alehegn Alemu**[1], **Molla Yigzaw Birhanu** [2], **Tebikew Yeneabat**[1,3], **Yoseph Merkeb Alamneh**[4], **Haile Amha**[5]

1 Department of Midwifery, College of Health Science, Debre Markos University, Debre Markos, Ethiopia, 2 College of Health Science, Debre Markos University, Debre Markos, Ethiopia, 3 Department of Midwifery, University of Technology Sydney, Sydney, Australia, 4 Department of Biomedical Sciences, School of Medicine, Debre Markos University, Debre Markos, Ethiopia, 5 Department of Nursing, College of Health Science, Debre Markos University, Debre Markos, Ethiopia

* melakd2018@gmail.com

**Data Availability Statement:** All relevant data are within the manuscript and its Supporting Information files.

## Abstract

### Background

Uterine rupture is the leading cause of maternal and perinatal morbidity and it accounts for 36% of the maternal mortality in Ethiopia. The maternal and perinatal outcomes of uterine rupture were inconclusive for the country. Therefore, this systematic review and meta-analysis aimed to estimate the pooled maternal and perinatal mortality and morbidity of uterine rupture and its association with prolonged duration of operation.

### Methods

The Preferred Reporting Items for Systematic Reviews and Meta-Analyses checklist was used for this systematic review and meta-analysis. We systematically used PubMed, Cochrane Library, and African Journals online databases for searching. The Newcastle-Ottawa quality assessment scale was used for critical appraisal. Egger's test and $I^2$ statistic used to assess the check for publication bias and heterogeneity. The random-effect model was used to estimate the pooled prevalence and odds ratios with 95% confidence interval (CI).

### Results

The pooled maternal mortality and morbidity due to uterine rupture in Ethiopia was 7.75% (95% CI: 4.14, 11.36) and 37.1% (95% CI: 8.44, 65.8), respectively. The highest maternal mortality occurred in Southern region (8.91%) and shock was the commonest maternal morbidity (24.43%) due to uterine rupture. The pooled perinatal death associated with uterine rupture was 86.1% (95% CI: 83.4, 89.9). The highest prevalence of perinatal death was

**Funding:** The author(s) received no specific funding for this work.

**Competing interests:** The authors have declared that no competing interests exist.

**Abbreviations:** ANC, Ante Natal Care; CI, Confidence interval; CS, Caesarean Section; DH, Demographic Health survey; LMICs, Low- and Middle-Income Countries; MMR, Maternal mortality ratio; OR, Odds Ratio; PRISMA, Preferred Reporting Items for Systematic Reviews and Meta-Analyses; SNNPR, Southern Nations, Nationalities and Peoples Representative; WHO, World Health Organization.

observed in Amhara region (91.36%) and the lowest occurred in Tigray region (78.25%). Prolonged duration of operation was a significant predictor of maternal morbidity (OR = 1.39; 95% CI: 1.06, 1.81).

## Conclusions

The percentage of maternal and perinatal deaths due to uterine rupture was high in Ethiopia. Uterine rupture was associated with maternal morbidity and prolonged duration of the operation was found to be associated with maternal morbidities. Therefore, birth preparedness and complication readiness plan, early referral and improving the duration of operation are recommended to improve maternal and perinatal outcomes of uterine rupture.

## Background

Uterine rupture is a rare obstetric complication and can be either complete or partial rupture [1, 2]. It is a life-threatening complications of pregnancy and childbirth in developing countries and is associated with high maternal and perinatal mortality and morbidity [3]. The World Health Organization (WHO) report also showed that uterine rupture is a serious obstetric complication being more prevalent and with more serious consequences in developing than developed countries [4]. In high-income countries, high burden uterine rupture occurs among women who attempt a trial of labour in subsequent pregnancy after previous caesarean section (CS) [5–7].

Uterine rupture is associated with adverse maternal outcomes like maternal near-miss, maternal mortality, and perinatal death. Maternal and perinatal mortality rates of uterine rupture were 8.7% and 97.8%, respectively [8]. In Ethiopia, maternal and perinatal mortality rates are still very high like that of other sub-Saharan African countries. The maternal mortality ratio (MMR) in Ethiopia was one of the highest, 412 per 100,000 live births and the perinatal mortality was 33 per 1000 live births in 2015 [9]. Uterine rupture is one of the major causes of maternal and perinatal mortalities in Ethiopia. Moreover, it accounts for 36% maternal mortality combined with obstructed labour and 58% of perinatal mortality [9].

It contributes for 2.7% of maternal deaths in Felege Hiwot Specialized hospital [10], 9.1% in Bench Maji zone [11], 11.1% in Adigrat town [5], and 21.4% of maternal death in Debre Markos Hospital [12]. The complications associated with uterine rupture includes severe anemia (24.4%), shock (40.7%), wound infection (16.7%), vesicovaginal fistula (12.5%), and sepsis (18.4%) [5].

Uterine rupture is managed with a surgical procedure called laparotomy. In attempts to improve overall patient safety during surgery and to reduce the prolonged duration of operation, WHO developed a checklist to be used at critical perioperative moments of induction, incision, and before leaving the operating room [13]. The use of the operation checklist reduced maternal morbidities associated prolonged duration of surgery [13]. There are several studies conducted on maternal and perinatal morbidities associated with uterine rupture and the effect of prolonged duration on maternal morbidities. However, these studies were not used an input for policy makers at the national level due to inconsistent and inconclusive findings. Thus, estimating the national level data is important. Therefore, this systematic review and meta-analysis was conducted to estimate the pooled prevalence of maternal and perinatal mortality and morbidity of uterine rupture. Also, this meta-analysis estimates the pooled effect of prolonged duration of operation on maternal morbidity of uterine rupture in Ethiopia.

## Methods

### Study design, setting and systematic review reporting

This systematic review and meta-analysis were designed to estimate the pooled prevalence of maternal and perinatal mortality and morbidity of uterine rupture and its association with prolonged duration of operation in Ethiopia. The protocol has been registered on an International Prospective Register of Systematic Review (PROSPERO), University of York Center for Reviews and Dissemination (https://www.crd.york.ac.uk/), registration number CRD42019119620. The findings of the review were reported based on the recommendation of the Preferred Reporting Items for Systematic Review and Meta-Analysis (PRISMA) guideline [14] (**S1 Checklist**).

### Data sources and search strategy

We systematically searched for the major international databases such as PubMed, Cochrane Library, Google Scholar, African Journals Online databases to find potentially eligible and relevant articles. The Google hand searching was used to retrieve studies. Besides, the search of the reference list of already identified studies to retrieve additional articles was done. The PECO (Population, Exposure, Comparison and Outcomes) search strategy was used for this review. Population included women who had uterine rupture in Ethiopia. Exposure was duration of the operation. The outcomes of this study were maternal mortality and maternal morbidity associated with uterine rupture and perinatal mortality due to uterine rupture in Ethiopia. The effect of prolonged duration of operation on maternal morbidities was also the outcome. Electronic databases were searched using keyword searching and the medical subject heading [MeSH] terms for each selected PECO component. The studies identified through systematic search were managed using Endnote X7. The keyword searching included "Maternal mortality OR maternal morbidity, perinatal mortality OR morbidity OR complications AND uterine rupture AND Ethiopia". The Boolean operators "OR" and "AND" were used to combine the searching terms (**S1 Table**).

### Eligibility criteria and study selection

The studies were included based on the following criteria, studies that reported maternal and perinatal mortality or morbidity of uterine rupture in Ethiopia, hospital based published studies in English language up to the end of our searching (30/6/2019) and cross-sectional, or case-control or cohort study design. Whereas, studies that were population case reports, surveillance data (Demographic and health survey), conference abstracts and articles that were not fully accessed after at least two email contacts of the principal investigator were excluded. Firstly, the two authors assessed the articles for inclusion through title, abstract and full paper review. Any disagreement was resolved by a consensus between the two reviewers. Then, potentially eligible studies underwent full-text review whether the predetermined set of criteria were met or not and for any duplicated records. Only the full-text article was retained when duplication was encountered [15].

### Outcome measurement and definitions

Maternal and perinatal mortality due to uterine rupture was the primary outcome of this review. Uterine rupture was defined as a partial or complete tear of the uterine wall during pregnancy or delivery [16]. Maternal morbidities and the effect of prolonged duration of operation on maternal morbidities was the secondary outcomes. The maternal morbidities of uterine rupture include sepsis, shock, severe anemia and fistula. Maternal mortality due to uterine rupture is defined as death of the mother from uterine rupture, its complications or management. Perinatal death of uterine rupture is defined as death of the fetus after 28 weeks of birth

to the first week of neonatal or early neonatal period from uterine rupture, its complications or management. Maternal morbidity is a defined when a woman who had uterine rupture develop at least one of the following conditions like sepsis, post-partum hemorrhage, anemia, relaparatomy, shock, pneumonia, organ failure, obstetric fistula, wound dehiscence, hysterectomy, and intensive care unit (ICU) admission.

Postpartum anemia was defined as low blood hemoglobin concentration, below $\leq$ 10 g/dl or hematocrit level less than $\leq$30% at 24 hours postpartum and severe anemia: if the below 9 g/dl or hematocrit level less than 27% at 24 hours postpartum. Prolonged duration of the operation is when the duration of operation took more than two hours [10].

WHO operative checklist: an operative safe surgery saves lives checklist consisting of 22-items to be used at critical perioperative moments of induction, incision, and before leaving the operating room [13]. Its main objectives include to operate on the correct patient at the correct site, to prevent harm from administration of anesthetics, while protecting the patient from pain, recognize and effectively prepare for life-threatening loss of airway or respiratory function, recognize and effectively prepare for risk of high blood loss, minimize the risk for surgical site infection, prevent inadvertent retention of instruments and sponges in surgical wounds, effectively communicate and exchange critical information for the safe conduct of the operationand enable Hospitals and public health systems will establish routine surveillance of surgical capacity, volume and results.

## Quality assessment and data collection

The quality of included studies was assessed using the Newcastle-Ottawa Scale quality assessment tool based on three components [17]. The primary component of the tool focused on the methodological quality of each study, graded from five stars. The second component was concerned about the comparability of each primary included study and the tool was graded from two stars. The last component of the tool used to assess the outcomes and statistical analysis of each original study, which was graded from three stars. Two authors independently assessed and extracted the articles for overall study quality using a standardized assessment tool and extraction format. The data extraction format included name of the primary author, publication year, region of the study, sample size, and prevalence of mortality and morbidities due to uterine rupture.

## Publication bias and statistical analysis

The publication bias was assessed using the Egger's tests [18] with a *p-value* less than *0.05*. $I^2$ test statistics was used to assess heterogeneity among studies, and a *p-value* less than *0.05* was used to declare heterogeneity. For the test result with the presence of heterogeneity, a random effect model was used as a method of analysis [9]. Data were extracted in Microsoft Excel and then exported to STATA version 11 for analysis. Subgroup analysis was conducted by region. Besides, a meta-regression model was done based on sample size and year of publication to identify the sources of random variations among included studies. The effect of prolonged duration of operation on maternal morbidity was analyzed using separate categories of meta-analysis. The findings of the meta-analysis were presented using forest plots and Odds Ratio (OR) with its 95% confidence interval (CI) [15].

## Results

### Study identification and characteristics of included studies

This systematic review and meta-analysis included published studies on the maternal and perinatal mortality and morbidities of uterine rupture in Ethiopia. The review found a total of 340

published articles. From those, 230 duplicated records were removed and 94 articles were excluded through screening of their title and abstracts. Finally, a total of 16 full-text papers were assessed for eligibility based on the prior inclusion and exclusion criteria. Of those 16, four studies were excluded due to lack of the outcome of interest and one study was excluded due to case report. Therefore, a total of 11 studies were included in the final meta-analysis (Fig 1).

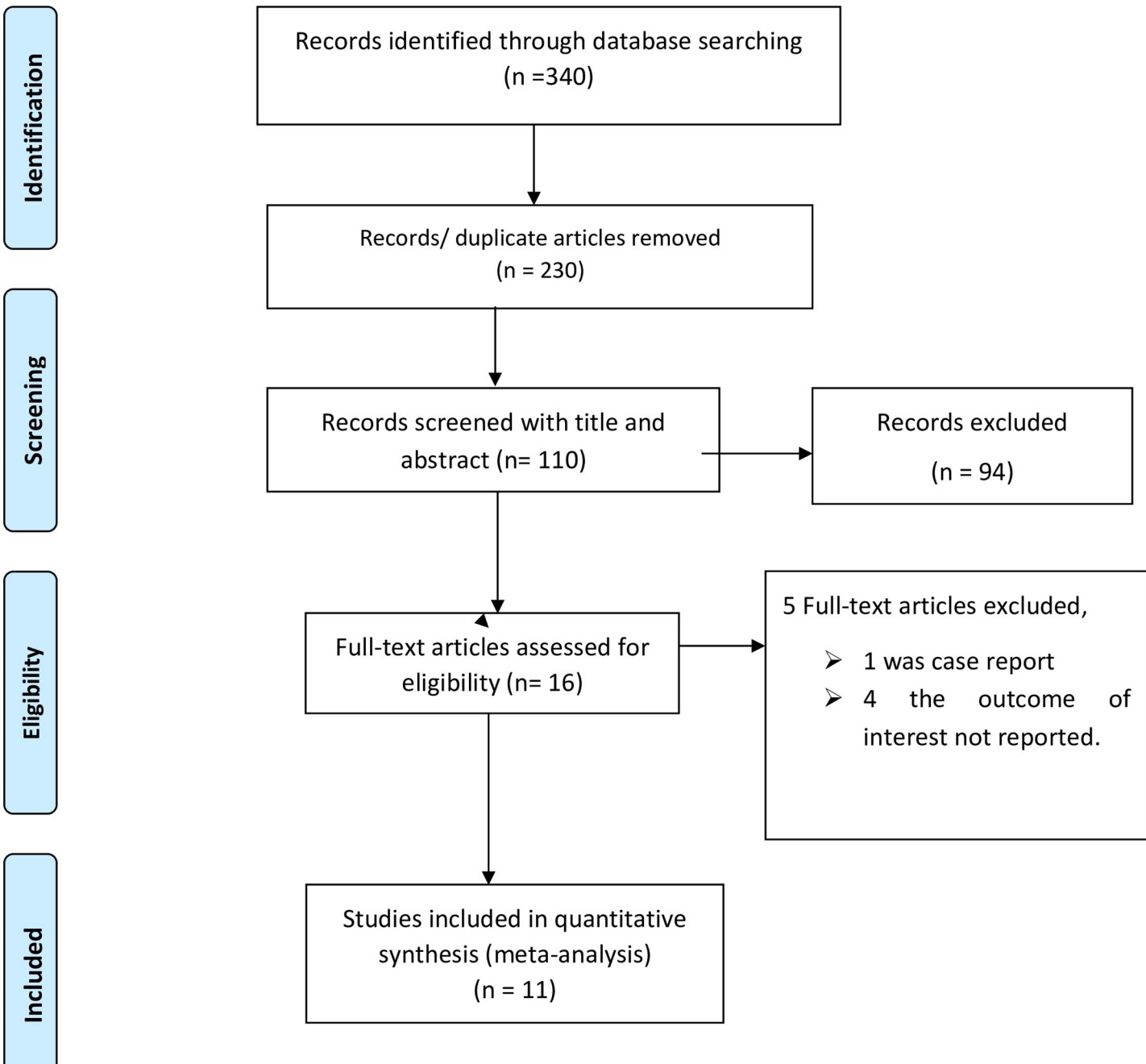

**Fig 1. PRISMA flow diagram of maternal and perinatal mortality of uterine rupture in Ethiopia.**

**Table 1. Characteristics of included studies in Ethiopia.**

| Author | Year | study design | region | sample | maternal mortality |
|---|---|---|---|---|---|
| Admasu A [12] | 2004 | cross sectional | Amhara | 1830 | 21.4 |
| Ahmed DM et al. [10] | 2018 | cross sectional | Amhara | 262 | 2.1 |
| Astatkie G et al. [20] | 2017 | cross sectional | Amhara | 242 | 6.6 |
| Bekabi TT [24] | 2018 | cross sectional | Oromia | 162 | 1.85 |
| Berhe Y et al [23] | 2015 | cross sectional | Tigray | 5185 | 2 |
| Dadi & Yanirbab [11] | 2017 | case control | SNNPR | 363 | 9.1 |
| Eshetie A et al. [22] | 2018 | cross sectional | SNNPR | 2498 | 8.7 |
| Gessesew [5] | 2002 | cross sectional | Tigray | 5980 | 11.1 |
| Mengstie H et al. [19] | 2018 | Cohort | SNNPR | 115 | 10.4 |
| Wossen A et al. | 2018 | cross sectional | Amhara | 312 | 2.7 |
| Yemane & Gizew [21] | 2016 | cross sectional | SNNPR | 352 | 9.9 |

Nine of the included studies were cross-sectional and the remaining two were case-control and cohort study designs. The highest sample size was 5,980 from a study conducted in Tigray region [5] and the minimum sample was 115 in a study conducted at Southern Nations, Nationalities and Peoples Region (SNNPR) [19]. Overall, the review was conducted among 17,301 women. All of the studies were conducted in five regions of the country. From these, four studies were from Amhara region [10, 12, 20], four were from SNNPR [11, 19, 21, 22], two were from Tigray [5, 23] and one study was from Oromia region [24] (Table 1).

## Prevalence of maternal and perinatal mortality due to uterine rupture

**Maternal mortality due to uterine rupture.** The meta-analysis of 11 studies showed the pooled prevalence of maternal mortality due to uterine rupture in Ethiopia was 7.75% (95% CI: 4.14, 11.36). A random-effects model of analysis was used due to a significant heterogeneity ($I^2$ = 98.8%, *p-value* < 0.05) (Fig 2). There was no publication bias based on the Eggers test (*p-value = 0.15*). Visual inspection of the funnel plot also showed symmetrical distribution of included articles (**S1 Table**). However, a univariate meta-regression analysis to identify the source of heterogeneity revealed a significant heterogeneity due to year of publication (*p-value = 0.006*) but non-significant due to the sample size (*p-value = 0.094*). The subgroup analysis also revealed that the highest prevalence of maternal mortality due to uterine rupture occurred in SNNPR (8.91% (95% CI: 7.91, 9.88)) and the lowest was in Tigray region (6.24% (95% CI: 2.37, 15.46)) (Table 2).

**Perinatal mortality due to uterine rupture.** The highest perinatal mortality has occurred in a study conducted in Debre Markos referral hospital, Amhara region (98.3%) [20] and was followed by a study conducted in Adigrat Hospital, Tigray region (98.1%) [5]. Whereas, the lowest magnitude of perinatal mortality was observed in a study conducted in Shire town Suhul General Hospital, Tigray region (35.3%) [25]. Based on the meta-analysis of 10 studies, the pooled prevalence of perinatal mortality among women with uterine rupture was 86.1% (95% CI: 83.4, 89.9) (Fig 3). The meta-analysis used the random effect model of analysis because of a significant heterogeneity. From the subgroup analysis, the prevalence of perinatal mortality was highest in Amhara region (91.36% (95% CI: 77.44, 99.8)) and the lowest occurred in Tigray region (78.25% (95% CI: 71.02, 85.48)) (Table 3). The funnel plot showed that there was symmetry (**S1 Fig**). The Duval Trim and Fill analysis was also conducted due to a significant publication bias (*p-value = 0.031*). A univariate meta-regression analysis also showed no significant variable responsible for the heterogeneity, year of publication (*p-value = 0.21*), sample size (*p-value = 0.42*), and type of study design (*p-value = 0.17*).

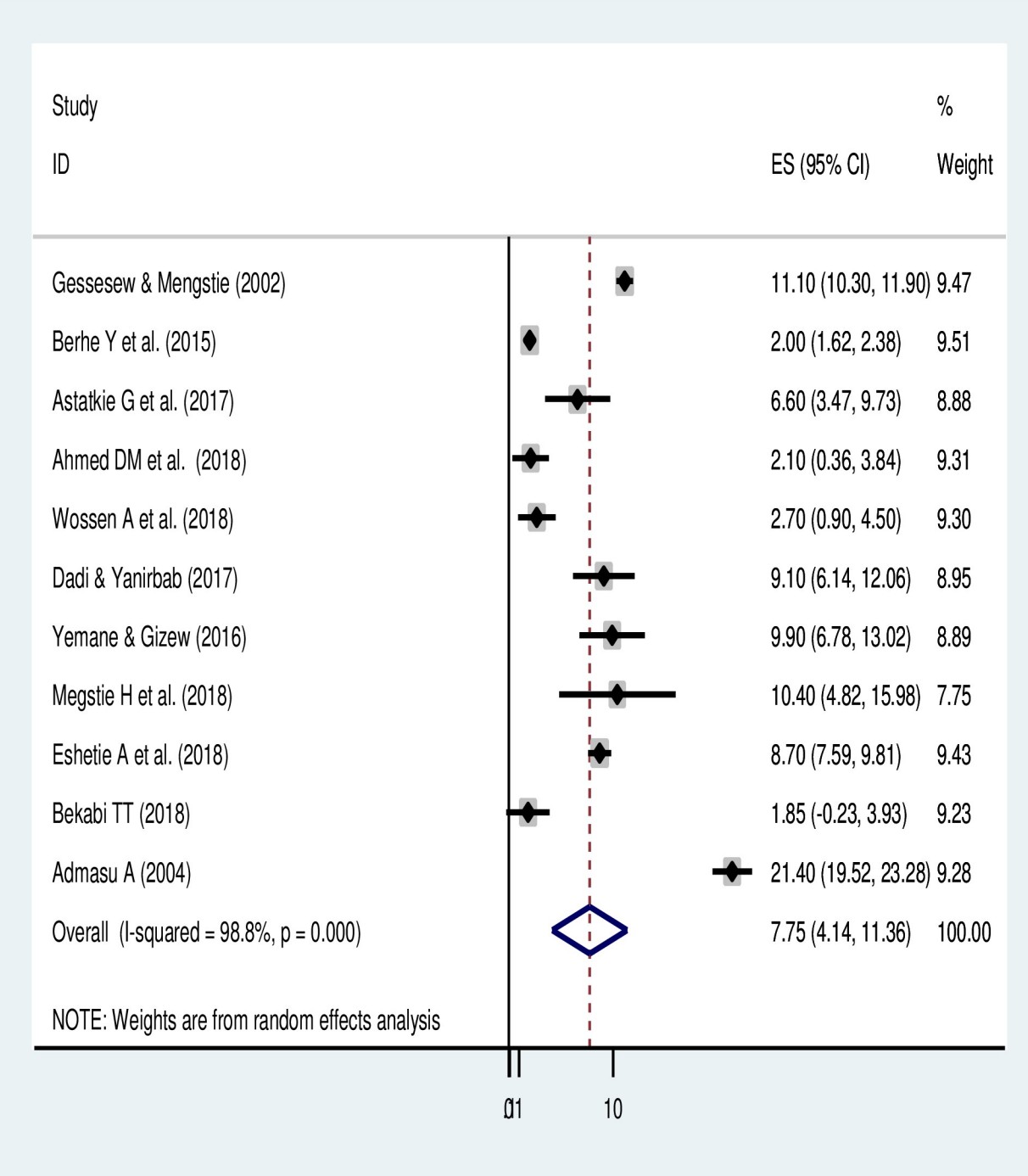

**Fig 2. Pooled maternal mortality of uterine rupture in Ethiopia.**

## Meta-analysis of maternal morbidities due to uterine rupture

The meta-analysis of 7 studies showed that 37.1% (95% CI: 8.44, 65.8) of women had a composite of at least one maternal complication associated with uterine rupture in Ethiopia. There was a significant heterogeneity among the included studies ($I^2$ = 99.9, *p-value = < 0.001*) (Fig 4). As the result a random-effects model of analysis was used (Table 4). The meta-analysis

**Table 2. Subgroup analysis on maternal mortality of uterine rupture by region in Ethiopia.**

| Region | Number of studies | ES [95%CI] | I² | P-value |
|---|---|---|---|---|
| SNNPR | 4 | 8.91 (7.91,9.88) | 75% | 0.00015 |
| Amhara | 4 | 8.2 (1.44,17.86) | 98.9% | 0.000001 |
| Tigray | 2 | 6.54 (2.37,15.46) | 99.8% | 0.0001 |

result of five studies revealed that the commonest maternal morbidities associated with uterine rupture were shock (24.43% (95% CI: 10.19, 38.67)) and sepsis (15.14% (95% CI: 9.52, 20.76)) (Fig 5 and Table 4).

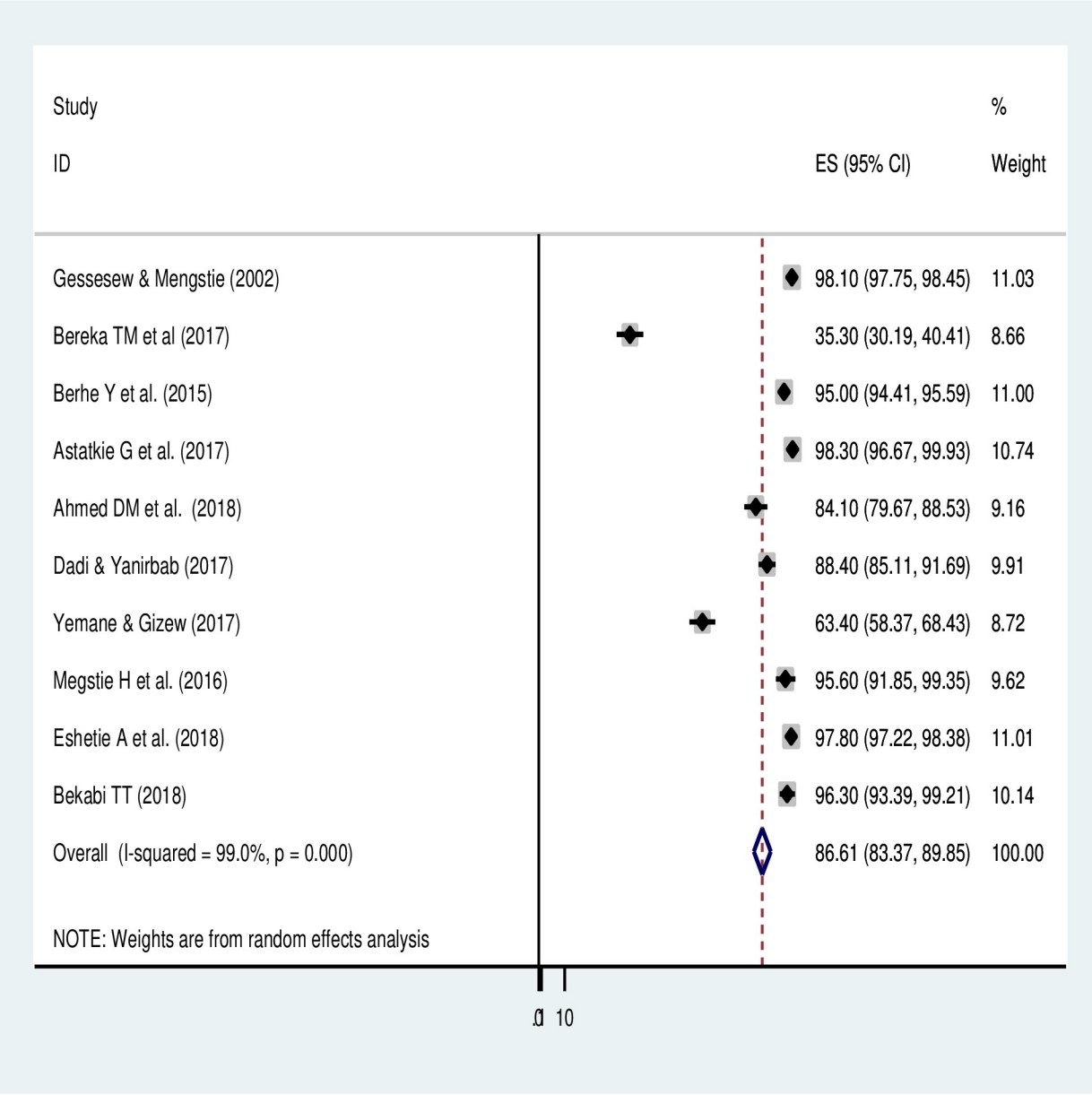

**Fig 3. Forest plot on prevalence of perinatal death of uterine rupture in Ethiopia.**

**Table 3. Subgroup analysis of perinatal death of uterine rupture by region.**

| Region | Prevalence (%) | No of studies | I² | P-value |
|---|---|---|---|---|
| Tigray | 78.25 (71.02, 85.48) | 3 | 99.7% | <0.0001 |
| SNNPR | 24.43 (10.19,38.67) | 4 | 98.5% | <0.0001 |
| Amhara | 91.36 (77.44, 99.8) | 2 | 97.1% | <0.0001 |

## Association between prolonged duration operation and maternal morbidity

The meta-analysis of 2 studies showed that prolonged duration of operation was a significant predictor of maternal morbidity among women with uterine rupture. Women who had prolonged duration of operation more than 2 hours had 1.39 times higher odds of maternal morbidity (OR:1.39; 95% CI: 1.06, 1.81) (Fig 6).

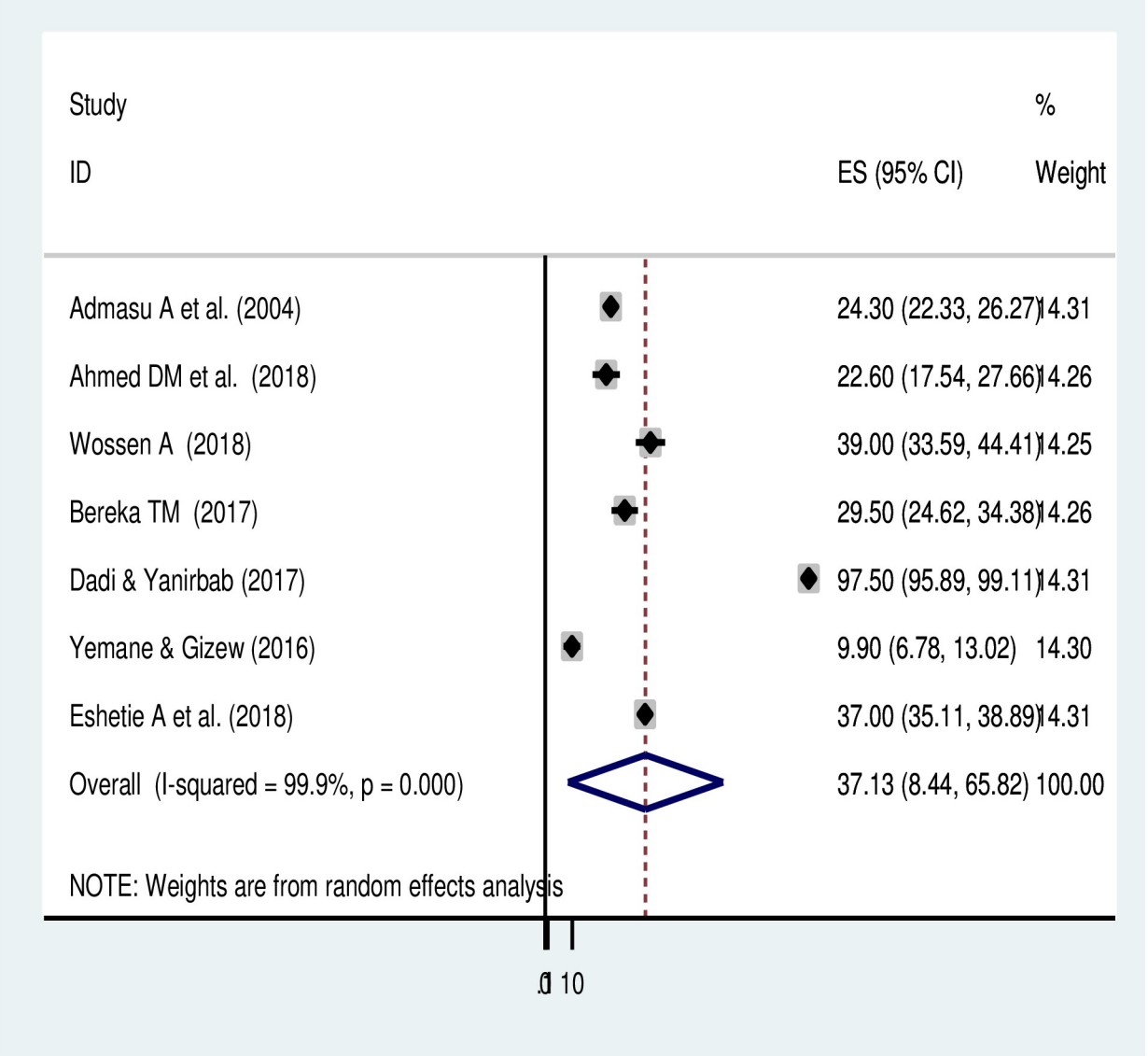

**Fig 4. Pooled prevalence of maternal morbidities of uterine rupture in Ethiopia.**

**Table 4. Type of maternal morbidities associated with uterine rupture in Ethiopia.**

| Maternal morbidities | Prevalence (95%CI) | No of studies | I² | P-value |
|---|---|---|---|---|
| Severe anemia | 13.48 (4.32,22.63) | 3 | 95% | <0.0001 |
| Shock | 24.43 (10.19,38.67) | 5 | 99.5% | <0.0001 |
| Sepsis | 15.14 (9.52,20.76) | 5 | 96.9% | <0.0001 |
| Fistula | 9.02 (6.04,11.99) | 5 | 90.3% | <0.0001 |

# Discussion

This study was conducted to estimate pooled prevalence of maternal and perinatal morbidity and mortality due to uterine rupture and its association with prolonged duration of operation

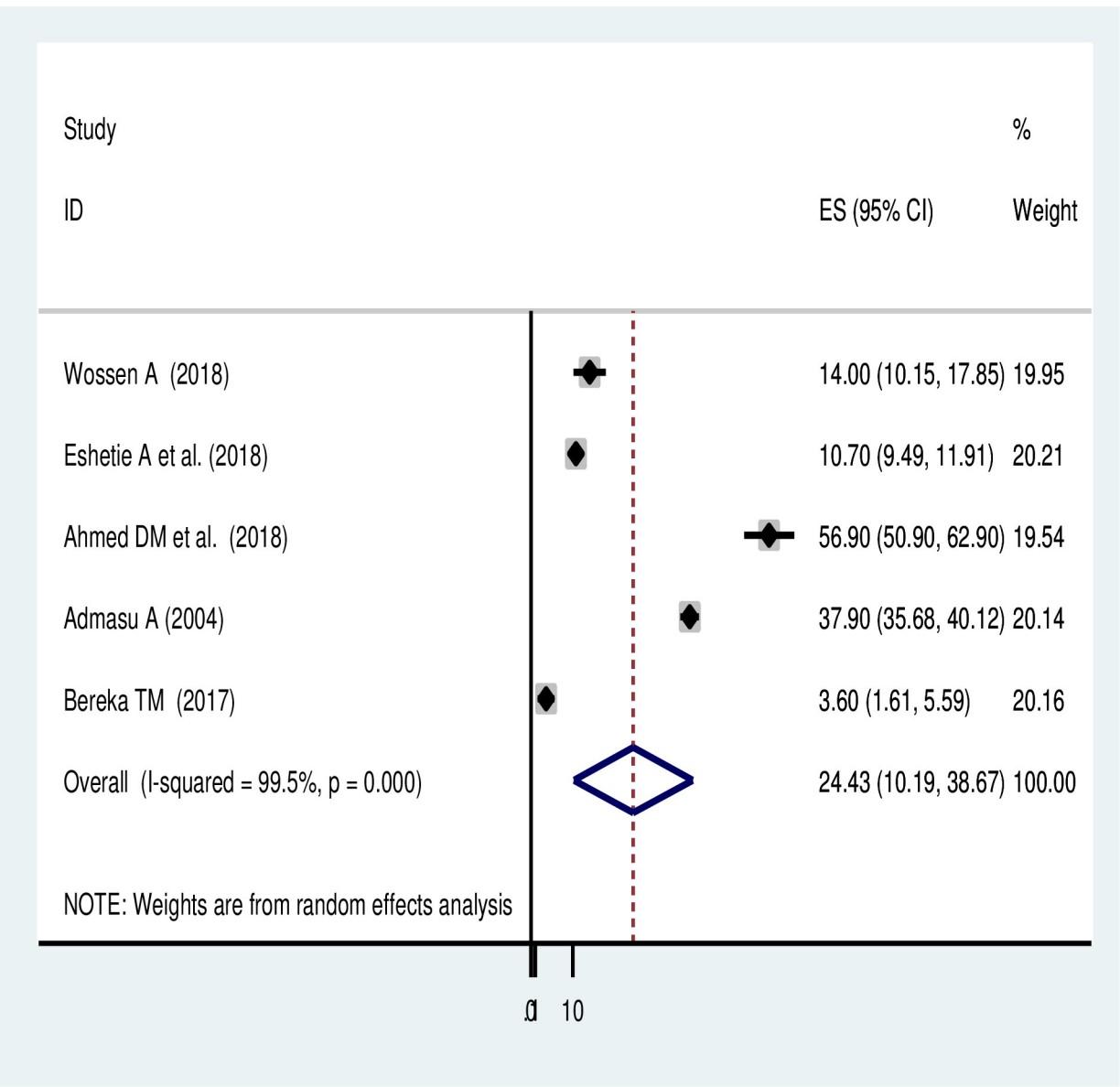

**Fig 5. Forest plot of shock associated with uterine rupture in Ethiopia.**

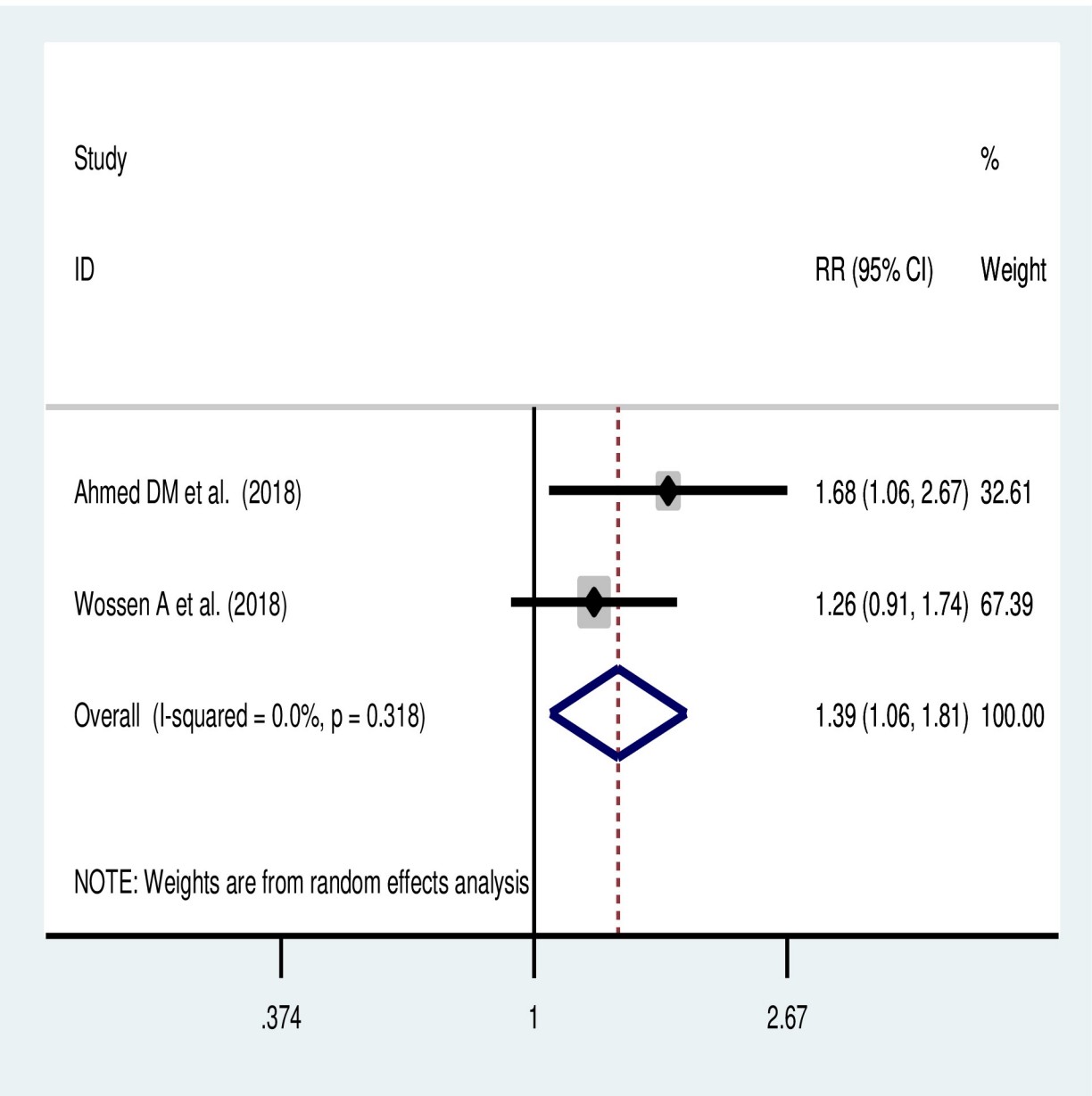

**Fig 6. The association of prolonged duration of operation and maternal morbidity.**

in Ethiopia. Accordingly, the pooled prevalence of maternal mortality associated with uterine rupture was 7.75% (95% CI: 4.14, 11.36). This was higher than 2.9% in a multi-country WHO survey report [8]. Whereas, a study conducted in the Netherlands reported no maternal death [12]. The possible difference might be explained due to the variation in the population characteristics, setting and quality of health care service provision and maternal health service utilization by the study participants. The high burden of obstructed labour and complications [26] resulting in metabolic disturbances, sepsis and shock in Ethiopia could also be the reasons for high percentage of maternal death. Moreover, previous studies also supported the evidence that obstructed labour, sepsis and anemia due haemorrhage are the leading causes of mortality associated with uterine rupture [27]. A study conducted in Debre Markos also showed that

37% of women with uterine rupture died from shock, 25% from severe anemia and 19% died from septic shock [20]. Thus, to reduce the high burden of maternal mortality due to uterine rupture, prevention of morbidities like haemorrhage, shock and sepsis can be accomplished through timely delivery by vacuum, forceps or caesarean section for prolonged labour with appropriate indications. Prenatally, anaemia can be prevented by Iron, Folic acid and vitamin supplementation. Protocols for induction and stimulation of labour also needs improvement.

The findings of this study was similar with studies done in Nigeria [28], 6.6% of maternal mortality associated with uterine rupture, but lower than the Senegal and Mali's report (12%) [29] and a study from Sudan (14.3%) [30]. The difference could be due to the difference in the percentage of women who gave birth at home, might be due to the high proportion of women in the area (almost 50%) of women gave birth at home, difference in the study period and a relative improvement of the healthcare system in the recent years in Ethiopia [9].

This systematic review and meta-analysis also revealed that more than one-third (37.1%; 95% CI: 8.44, 65.8) of women had maternal morbidities associated with uterine rupture. This finding is in line with a 31.2% from a global study conducted in 29 counties [8]. The findings of this meta-analysis also found that the highest prevalence of maternal mortality associated with uterine rupture has occurred in SNNPR and the lowest was in Tigray region. The possible reason for the regional variation in the level of maternal morbidities associated with uterine rupture in Ethiopia could be explained to the difference in maternal health care service utilization. Hence, a national demographic health survey report in Ethiopia supported that the lowest and highest antenatal care (ANC) service utilization were spatially clustered in SNNPR (39.8%) and Tigray region (65%) next to Addis Ababa (94%), respectively [31]. Hence, a systematic review support that ANC visit significantly increase birth in the health facility [32].

This systematic review and meta-analysis study also found that more than three-fourths (86.1%) of perinatal death in Ethiopia were due to uterine rupture. The finding was higher than studies done in 29 countries by WHO (49.4%) [8], a review study (61%) [33] and a study conducted in Nigeria (73%) [28]. However the finding was lower than a 92.8% from a study conducted in Sudan [30]. This difference could be due to the fact that women who had uterine rupture in Ethiopia might not get skilled birth attendant earlier and within the golden time due to the poor decisions about when to seek care during childbirth [10, 11, 34]. Furthermore, delay in getting care is associated with increased risk of uterine rupture and increases the risk of fetal mortality. It might also be due to the lower level of maternal health service utilization, low level of birth preparedness and complication readiness plan and delay of getting emergency obstetric care [22, 34, 35].

Additionally, studies supported that having severe maternal near-miss was associated with adverse perinatal outcomes [36–38]. The findings from a study conducted in Nigeria showed a larger proportion of women (97.8%) who had uterine rupture were provided with obstetric interventions in labour and it reduce the burden of perinatal mortality. Also, the high proportion of obstetrical fistula might be associated with the high burden of perinatal mortality as supported by recent studies. Fistula increased the odds of stillbirth or perinatal death [39]. A meta-analysis also revealed that 90.1% of women who developed fistula resulted in stillbirth or the risk of stillbirth was 99 times greater among women with fistula [40]. Therefore, basic surgical care through an improvement of the comprehensive emergency obstetric and newborn care without delay is mandatory to reduce perinatal death and obstetric fistula among women [41].

Likewise, this systematic review and meta-analysis revealed that prolonged duration of operation was significantly associated with maternal morbidity among women who had uterine rupture. Women who had prolonged duration of operation were 39% more likely to had maternal morbidities than their counterparts. This finding is in accordance with other study [10]. Prolonged duration of operation is a significant public health issue with a great disparity

in low- and middle-income countries (LMICs). Barriers such as accessibility, availability, affordability and acceptability of surgical care may affect the outcomes of operation in Ethiopia [42] and LMICs [43–48]. For this, evidence suggests that sustainable WHO surgical safety checklist implementations to improve surgical care in these settings can be a cost-effective intervention to reduce maternal morbidity through safe surgery [49]. However, successful WHO surgical safety checklist implementation in Africa is largely limited to single center implementations without longitudinal evaluation of sustainability [42, 50–52], which is also similar in Ethiopia [42]. Thus, adherence to the WHO surgical safety checklist to ensure patient safety during surgery should be improved to reduce morbidity and mortality after surgery [53, 54]. Moreover, a systematic review and meta-analysis supported that adherence of the WHO surgical safety checklist reduces any post-operative complications by 41% and mortality by 23% [54]. Thus, the government should go in accordance with the Safe surgery plan like scaleup surgical capacities [55]. The duration of operation is dependent on the type and extent of rupture, hemodynamic status of the mother, desire for future fertility, the type of anesthesia used, level of health care, availability of equipment and consumables, presence of gross infection and experience of the surgeon.

This study was the first systematic review and meta-analysis in Ethiopia, and the included studies show the causal relationship. However, the findings of this review should be interpreted with some limitation. The high heterogeneity of results among studies could be explained by the heterogeneity in the characteristics of the studies such as year of publication and this may have led to insufficient power to detect statistically significant association. The studies were conducted only in the five regions of the country, and this may reduce its representativeness. Some of the included studies also had a small sample size, and this may might affect the estimation of the results. This study also can't assess the effect of prolonged duration of operation, women having uterine rupture with bladder rupture and type of uterine rupture on maternal and perinatal mortality due to absence of data on the included studies.

## Conclusions

Uterine rupture was associated with maternal and perinatal morbidity, and the percentage of maternal deaths was high in Ethiopia. Prolonged duration of operation was found to be associated with maternal morbidities due to uterine rupture. Therefore, birth preparedness and complication readiness plan, early referral and improving the duration of operation with the WHO standards are recommended to reduce maternal mortality and morbidity and stillbirth among women with uterine rupture. Future prospective studies should assess the effect of duration of operation and type of uterine rupture on maternal and perinatal mortality.

## Supporting information

**S1 Checklist. PRISMA checklist: Maternal and perinatal mortality of uterine rupture.**
(DOC)

**S1 Fig. Forest plot of prevalence of maternal mortality due to uterine rupture.**
(TIF)

**S2 Fig. Forest plot of perinatal death associated with uterine rupture.**
(TIF)

**S1 Table. PubMed search of maternal and perinatal mortality and morbidity of uterine rupture.**
(DOCX)

## Author Contributions

**Conceptualization:** Melaku Desta, Tebikew Yeneabat, Haile Amha.

**Data curation:** Melaku Desta, Getachew Mullu Kassa, Yewbmirt Sharew, Addisu Alehegn Alemu, Yoseph Merkeb Alamneh, Haile Amha.

**Formal analysis:** Melaku Desta.

**Funding acquisition:** Melaku Desta.

**Investigation:** Melaku Desta, Getachew Mullu Kassa, Haile Amha.

**Methodology:** Melaku Desta, Getachew Mullu Kassa, Yewbmirt Sharew, Haile Amha.

**Project administration:** Getachew Mullu Kassa, Yewbmirt Sharew, Addisu Alehegn Alemu.

**Resources:** Haile Amha.

**Software:** Melaku Desta.

**Supervision:** Getachew Mullu Kassa, Addisu Alehegn Alemu, Yoseph Merkeb Alamneh.

**Validation:** Melaku Desta, Getachew Mullu Kassa, Addisu Alehegn Alemu.

**Visualization:** Melaku Desta, Getachew Mullu Kassa, Tebikew Yeneabat, Yoseph Merkeb Alamneh.

**Writing – original draft:** Melaku Desta, Haile Amha.

**Writing – review & editing:** Melaku Desta, Getachew Mullu Kassa, Temesgen Getaneh, Molla Yigzaw Birhanu, Tebikew Yeneabat, Yoseph Merkeb Alamneh.

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
