## [Decision Letter · Decision Letter 0]

18 Aug 2020

PONE-D-20-12283

Maternal and perinatal mortality and morbidity of uterine rupture and its association with prolonged duration of operation in Ethiopia: a systematic review and meta-analysis

PLOS ONE

Dear Dr. desta,

Thank you for submitting your manuscript to PLOS ONE. After careful consideration, we feel that it has merit but does not fully meet PLOS ONE’s publication criteria as it currently stands. Therefore, we invite you to submit a revised version of the manuscript that addresses the points raised during the review process.

Thanks for submitting on a relevant topic. The methods used is very good but communication of the main messages can be improved significantly. 

We look forward to receiving your revised manuscript.

Kind regards,

Charles A. Ameh, PhD, MPH, FWACS (OBGYN), FRCOG

Academic Editor

PLOS ONE

Additional Editor Comments:

Thanks for submitting on a relevant topic. The methods used is very good but communication of the main messages can be improved significantly. I will recommend that a very good proof reading english language expert is engaged before resubmission to improve the grammar, typographic errors and messaging. Specifically the conclusions in the abstract are not consistent with the results. The main outcome measured is the maternal and perinatal mortality ASSOCIATED with ruptured uterus, this is what I understood after the reading the manuscript but this is not what is communicated in the methods section (outcome measurement). Both reviewers have also pointed out issues with grammar and messaging. Please update the abstract with the result/conclusion of the critical appraisal, test for heterogenicity and publication bias.

2. We noticed you have some minor occurrence of overlapping text with previous publications, which needs to be addressed.

In your revision ensure you cite all your sources (including your own works), and quote or rephrase any duplicated text outside the methods section.

Further consideration is dependent on these concerns being addressed.

3. Please provide the complete search strategy for at least one database as a new supporting information file.

4. Please amend the manuscript submission data (via Edit Submission) to include author Getachew Mullu Kassa.

5. Please amend your authorship list in your manuscript file to include author Getachew Mullu desta.

Reviewers' comments:

Reviewer's Responses to Questions

**Comments to the Author**

1. Is the manuscript technically sound, and do the data support the conclusions?

Reviewer #1: Yes

Reviewer #2: No

2. Has the statistical analysis been performed appropriately and rigorously? 

Reviewer #1: Yes

Reviewer #2: No

3. Have the authors made all data underlying the findings in their manuscript fully available?

Reviewer #1: Yes

Reviewer #2: Yes

4. Is the manuscript presented in an intelligible fashion and written in standard English?

Reviewer #1: Yes

Reviewer #2: No

5. Review Comments to the Author

Reviewer #1: On a general note the whole of the discussion section needs to be reviewed due to a lot of grammatical errors which clouds the intended messages. This article can be considered for publication if these corrections are made. Thank you.

Reviewer #2: Thanks for the opportunity to review the manuscript. And I applaud the authors for working on such an important subject.

I have the following observations for consideration.

1. Aim and title: This need revising. The title seems to suggest that the study aims to find out the association between morbidity/mortality and duration of operation. However, the results section deliberates more on analysis of maternal and perinatal outcomes of the 11 studies and very little about the effect of duration of operation. This makes it difficult to decipher the key message from the manuscript. Is the study aiming at evaluating compliance to the 22 items in the WHO checklist? Is it assessing the association between duration of operation and maternal/perinatal outcomes? How will this help improve maternal and perinatal care in Ethiopia?

2. Design: The study identifies more with a literature review and meta-analysis of data than a systematic review. Within the PECO framework chosen, the exposure suggested is “duration of operation” however performing an obstetric operation involves other items that may confound the duration measured. This includes the nature and extend of the uterine rupture, the type of anesthesia used, level of health care, availability of equipment and consumables. A statement acknowledging possibility of such confounders would be helpful.

3. Key terms used: Terms such as morbidity, shock and anaemia may need some more clarification to improve understanding. For example, shock is reported as a leading cause of morbidity. It remains to clarify how morbidity was defined in this study and in the studies included in the review. It will also be helpful to know when complications such as anaemia were diagnosed. How long after the uterine rupture? A mention of if complications were outlined as immediate and long term in the studies and if they were presented separately.

4. Causal effect: The paragraph just before the conclusion section asserts that this meta-analysis has shown a causal relationship. This is a strong statement that needs clarity into how it was reached. Similarly the statement…”and an improvement of the health care system is our country due to the commitment of the government to reduce maternal mortality with increased qualified staffs” may need rephrasing or evidence for the same.

5. Grammar and typos need attention throughout the document. No page numbers indicated.

6. PLOS authors have the option to publish the peer review history of their article (what does this mean?). If published, this will include your full peer review and any attached files.

Reviewer #1: **Yes: **PROF HAJARATU UMAR SULAYMAN

Reviewer #2: No

---

## [Author Response · Author response to Decision Letter 0]

23 Sep 2020

Dear reviewer, there has been a major revision of the whole structure of the manuscript (Abstract, introduction, methods, results, discussion and conclusions) mainly with a correction of grammar errors. The journal requirements supposed by you have been included as search strategy has included as supplementary file and the whole structure of the manuscript has been revised to be easy for readers. We hope now the manuscript is clear and more acceptable than its previous version. We tried to reduce the overlap with previous publications and we also cite the sources in the methods section. We have tried to present the response for each reviewer according to your comment what to supposed to do so. For this, here we have given our responses to each of the concerns you raised, highlighted by red color. Again, we would like to remind our strongest gratitude for your effort for the improvement of this manuscript and the response for each the points were addressed in the response to reviewers’ section.

Regards

---

## [Decision Letter · Decision Letter 1]

23 Dec 2020

PONE-D-20-12283R1

Maternal and perinatal mortality and morbidity of uterine rupture and its association with prolonged duration of operation in Ethiopia: a systematic review and meta-analysis

PLOS ONE

Dear Dr. Desta,

Thank you for submitting your manuscript to PLOS ONE. After careful consideration, we feel that it has merit but does not fully meet PLOS ONE’s publication criteria as it currently stands. Therefore, we invite you to submit a revised version of the manuscript that addresses the points raised during the review process.

The manuscript and the reviewers’ comments were carefully evaluated. The Reviewers appreciated the manuscript. Nevertheless, as suggested, the manuscript requires some improvement before to be considered for publication. Suggested revisions are in detail reported in the Reviewers’ comments.

We look forward to receiving your revised manuscript.

Kind regards,

Simone Garzon

Academic Editor

PLOS ONE

Reviewers' comments:

Reviewer's Responses to Questions

**Comments to the Author**

1. If the authors have adequately addressed your comments raised in a previous round of review and you feel that this manuscript is now acceptable for publication, you may indicate that here to bypass the “Comments to the Author” section, enter your conflict of interest statement in the “Confidential to Editor” section, and submit your "Accept" recommendation.

Reviewer #1: All comments have been addressed

Reviewer #2: (No Response)

Reviewer #3: (No Response)

2. Is the manuscript technically sound, and do the data support the conclusions?

Reviewer #1: Yes

Reviewer #2: Yes

Reviewer #3: Yes

3. Has the statistical analysis been performed appropriately and rigorously? 

Reviewer #1: Yes

Reviewer #2: Yes

Reviewer #3: Yes

4. Have the authors made all data underlying the findings in their manuscript fully available?

Reviewer #1: Yes

Reviewer #2: Yes

Reviewer #3: Yes

5. Is the manuscript presented in an intelligible fashion and written in standard English?

Reviewer #1: Yes

Reviewer #2: No

Reviewer #3: Yes

6. Review Comments to the Author

Reviewer #1: The author has addressed all the concerns raised and the grammatical errors have been corrected. He also rephrased some of the sentences that were not clear.

Reviewer #2: Thanks for addressing many of the comments given during the first review. However, I am afraid, there is more to do in terms of use of technical terms and grammatical correctness.

In the methods section (Page 4), I don't think you need to present the basic information about Ethiopia. You could safely delete the second and third sentences.

It will be helpful for the conclusion to be clear whether increased duration of operation is associated with increased severity or number of complications. This is not yet clear.

There is inconsistent use of the terms stillbirth, perinatal deaths, neonatal deaths eg page 3. Please check and revise. A stillbirth is also a perinatal death. The fourth statement on page 5: ... "Perinatal death of uterine rupture

is defined as death of the fetus after 28 weeks of birth to the first week of neonatal or early neonatal

period from uterine rupture, its complications or management". is not clear.

Page 7: Not mentioned how you ended up with 11 studies from 16, although the information is available on the PRISMA diagram.

Some final grammatical considerations:

Examples:

Page 3: Third paragraph... complication instead of complications.

Page 5: ... a hospital based published studies. "a" is not required.

Please double check throughout the manuscript.

Thanks.

Reviewer #3: Title: Maternal and perinatal mortality and morbidity of uterine rupture and its association with prolonged duration of operation in Ethiopia: a systematic review and meta-analysis

Comments :

1. Abstract :

a. As a whole includes 368 words while in the journal regulation should not exceed 300!!!!

b. The background is too long and away from your objectives i.e The adverse outcomes of uterine rupture were highly variable and inconclusive across different studies in the country is recorded in the background and your objectives was not to find out the true adverse outcomes !!!! What were your adverse out comes???

c. Specifying the effect of increased maternal and perinatal outcomes after UR just by (prolonged duration of operation in Ethiopia) is not justifiable!!! As you have to elude other causes for increasing this rate for instance delayed of presentation to the hospital! The woman was in collapse state at time of admission that u failed to resuscitate her and the woman was died not because of the prolonged operation time? The women had heart disease with RU!!!! The operation was prolonged because she had rupture bladder with the RU!!! may be

d. Conclusion: Uterine rupture was associated with maternal morbidity and prolonged duration of the operation was found to be associated with maternal morbidities : Again u have to clarify this statement in the methodology

e. The Preferred Reporting Items for Systematic Reviews and Meta-Analyses (PRISMA) mentioned once in abstract, no need to add the abbreviation here.

f. Therefore, birth preparedness and complication readiness plan, early referral and improving the duration of operation according to the World Health Organization standards are recommended: that is true all these factors involves to increase morbidity and mortality but the problem u haven’t identify these factors in your methodology or result to compare it with the duration of operation.

g. to reduce maternal morbidity and mortality and stillbirth among women with uterine rupture: Please unify your results regarding what do you looked for in the article in relation to fetal or neonatal out comes .Was it just stillbirth rate or perinatal outcome? , or admission to NICU?

2. Introduction:

a. WHO developed a checklist consisting of 22-items to be used at critical perioperative moments of induction, incision, and before leaving the operating room : These check lists should be identified in detail in methods section

Outcome measurement:

1. Change the title to : Outcome measurements and definitions

2. Second line: Uterine rupture is a partial…… uterine rupture was defined as ….

3. U have to define all your variables here , Maternal mortality, Morbidity , perinatal death, stillbirth, sepsis ,severe anemia Fistula (which type , shock, …..and all should have references

4. Maternal morbidities of uterine rupture include sepsis, shock, severe anemia and fistula: Maternal morbidities after uterine rupture include much more morbidities which were not included in this study!!! u have to clarify u haven’t identify them and why u have only recorded (sepsis, fistula, shock and severe anemia).

5. How long women after operation were followed up to confirm Fistula and sepsis?

6. Are there any women died before the operation was conducted for them??? And did they included in the analysis or excluded?

Result:

1. Screening of their title and abstracts. Finally, a total of 16 full-text papers were assessed for eligibility based on the prior inclusion and exclusion criteria. Therefore, a total of 11 studies were included in the final meta-analysis: why 11 out of 16?? please identify the cause

7. PLOS authors have the option to publish the peer review history of their article (what does this mean?). If published, this will include your full peer review and any attached files.

Reviewer #1: **Yes: **PROFESSOR HAJARATU UMAR SULAYMAN

Reviewer #2: No

Reviewer #3: **Yes: **Professor. Shahla kareem Alalaf

---

## [Decision Letter · Decision Letter 2]

12 Jan 2021

Maternal and perinatal mortality and morbidity of uterine rupture and its association with prolonged duration of operation in Ethiopia: a systematic review and meta-analysis

PONE-D-20-12283R2

Dear Dr. Desta,

We’re pleased to inform you that your manuscript has been judged scientifically suitable for publication and will be formally accepted for publication once it meets all outstanding technical requirements.

Kind regards,

Simone Garzon

Academic Editor

PLOS ONE

Additional Editor Comments (optional):

Reviewers' comments:

Reviewer's Responses to Questions

**Comments to the Author**

1. If the authors have adequately addressed your comments raised in a previous round of review and you feel that this manuscript is now acceptable for publication, you may indicate that here to bypass the “Comments to the Author” section, enter your conflict of interest statement in the “Confidential to Editor” section, and submit your "Accept" recommendation.

Reviewer #2: All comments have been addressed

Reviewer #3: All comments have been addressed

2. Is the manuscript technically sound, and do the data support the conclusions?

Reviewer #2: Yes

Reviewer #3: Yes

3. Has the statistical analysis been performed appropriately and rigorously? 

Reviewer #2: Yes

Reviewer #3: Yes

4. Have the authors made all data underlying the findings in their manuscript fully available?

Reviewer #2: Yes

Reviewer #3: Yes

5. Is the manuscript presented in an intelligible fashion and written in standard English?

Reviewer #2: Yes

Reviewer #3: Yes

6. Review Comments to the Author

Reviewer #2: Thank you for working hard on the manuscript. It is clear that adequate attention and effort were put in reviewing the manuscript.

I have only two minor grammatical suggestions to make:

1. Page 9, 2nd paragraph on the discussion: The first sentence reads, "The findings of this study was similar with.." you may consider rephrasing this to read, "Our findings are similar with..."

2. Page 11, paragraph before Conclusion. Last but one sentence. It reads... "this may might affect..." please consider revising as one word could be redundant.

Thanks.

Reviewer #3: -The comments required were addressed sufficiently by the author

-Please be aware that the style of the references in the text to be uniform

7. PLOS authors have the option to publish the peer review history of their article (what does this mean?). If published, this will include your full peer review and any attached files.

Reviewer #2: **Yes: **Mselenge Mdegela

Reviewer #3: **Yes: **Shahla Kareem Alalaf

---

## [Editor Report · Acceptance letter]

12 Apr 2021

PONE-D-20-12283R2 

Maternal and perinatal mortality and morbidity of uterine rupture and its association with prolonged duration of operation in Ethiopia: a systematic review and meta-analysis   

Dear Dr. Desta:

I'm pleased to inform you that your manuscript has been deemed suitable for publication in PLOS ONE. Congratulations! Your manuscript is now with our production department. 

Kind regards, 

on behalf of

Dr. Simone Garzon 

Academic Editor

PLOS ONE